# Complex Modes of Inheritance in Hereditary Red Blood Cell Disorders: A Case Series Study of 155 Patients

**DOI:** 10.3390/genes12070958

**Published:** 2021-06-23

**Authors:** Immacolata Andolfo, Stefania Martone, Barbara Eleni Rosato, Roberta Marra, Antonella Gambale, Gian Luca Forni, Valeria Pinto, Magnus Göransson, Vasiliki Papadopoulou, Mathilde Gavillet, Mohsen Elalfy, Antonella Panarelli, Giovanna Tomaiuolo, Achille Iolascon, Roberta Russo

**Affiliations:** 1Dipartimento di Medicina Molecolare e Biotecnologie Mediche, Università degli Studi di Napoli Federico II, 80131 Napoli, Italy; andolfo@ceinge.unina.it (I.A.); stefaniamartone85@libero.it (S.M.); rosato.barbara@gmail.com (B.E.R.); robertamarra.r@gmail.com (R.M.); roberta.russo@unina.it (R.R.); 2CEINGE Biotecnologie Avanzate, 80145 Naples, Italy; antonellagambale@gmail.com (A.G.); panarelli@ceinge.unina.it (A.P.); g.tomaiuolo@unina.it (G.T.); 3Department of Laboratory Medicine (DAIMedLab), UOC Medical Genetics, ‘Federico II’ University Hospital, 80131 Naples, Italy; 4Centro della Microcitemia e delle Anemie Congenite, Ospedale Galliera, 16128 Genoa, Italy; gianluca.forni@galliera.it (G.L.F.); valeria.pinto@galliera.it (V.P.); 5Department of Paediatrics, The Queen Silvia Children’s Hospital, Sahlgrenska University Hospital, 41345 Gothenburg, Sweden; magnus.l.goransson@vgregion.se; 6Service and Central Laboratory of Haematology, Department of Oncology and Department of Laboratory Medicine and Pathology, Lausanne University Hospital (CHUV), 1011 Lausanne, Switzerland; vasiliki.papadopoulou@chuv.ch (V.P.); mathilde.gavillet@chuv.ch (M.G.); 7Thalassemia Centre, Faculty of Medicine, Ain Shams University, Cairo 11566, Egypt; elalfym@hotmail.com; 8Department of Chemical Engineering, Materials and Industrial Production, ‘Federico II’ University of Naples, 80125 Naples, Italy

**Keywords:** red blood cell defects, targeted next-generation sequencing, multi-locus inheritance, *PIEZO1*, *SPTA1*

## Abstract

Hereditary erythrocytes disorders include a large group of conditions with heterogeneous molecular bases and phenotypes. We analyzed here a case series of 155 consecutive patients with clinical suspicion of hereditary erythrocyte defects referred to the Medical Genetics Unit from 2018 to 2020. All of the cases followed a diagnostic workflow based on a targeted next-generation sequencing panel of 86 genes causative of hereditary red blood cell defects. We obtained an overall diagnostic yield of 84% of the tested patients. Monogenic inheritance was seen for 69% (107/155), and multi-locus inheritance for 15% (23/155). *PIEZO1* and *SPTA1* were the most mutated loci. Accordingly, 16/23 patients with multi-locus inheritance showed dual molecular diagnosis of dehydrated hereditary stomatocytosis/xerocytosis and hereditary spherocytosis. These dual inheritance cases were fully characterized and were clinically indistinguishable from patients with hereditary spherocytosis. Additionally, their ektacytometry curves highlighted alterations of dual inheritance patients compared to both dehydrated hereditary stomatocytosis and hereditary spherocytosis. Our findings expand the genotypic spectrum of red blood cell disorders and indicate that multi-locus inheritance should be considered for analysis and counseling of these patients. Of note, the genetic testing was crucial for diagnosis of patients with a complex mode of inheritance.

## 1. Introduction

Hereditary anemias are a heterogeneous group of conditions that are characterized by complex genotype–phenotype correlations. Based on clinical manifestations and morphological red blood cell (RBC) alterations, hereditary anemias can be broadly classified into four different subtypes: (i) disorders of hemoglobin synthesis, such as thalassemia and hemoglobinopathies; (ii) hypo-regenerative anemias, such as congenital dyserythropoietic anemias; (iii) aregenerative anemias, such as Diamond–Blackfan anemia (iv) RBC membrane defects that are due to either alterations to the structural organization of the membranes, such as hereditary spherocytosis (HS), hereditary pyropoikilocytosis (HPP), and hereditary elliptocytosis, or to alterations to membrane transport functions, such as hereditary stomatocytosis; and (v) nonspherocytic hemolytic anemias, due to RBC enzyme defects [1,2,3,4,5,6,7].

The conventional workflow for diagnosis of hereditary anemias starts with the first line of investigation of evaluation of familial history, with complete blood counts and peripheral blood smears. Then, specialized biochemical tests are required. Finally, genetic testing serves as the confirmatory test. Currently, genetic testing is used early in the diagnostic workflow of hereditary anemias, which removes the need for some of the specialized tests [8,9], especially when the clinical data for the patients are not informative, or when the patient is transfusion dependent. Next-generation sequencing (NGS), as mainly the targeted NGS approach, has revolutionized the framework of the diagnosis of hereditary anemias by reducing both time and cost. Nevertheless, it remains a challenge to diagnose many hereditary anemia phenotypes according to phenotypic features and conventional diagnostic testing. In general clinical genetics setting, the diagnostic yield ranges from 38% to 87% of patients, depending on how many and which genes are included, and on the depth of the phenotypic assessment required [9]. A drawback of NGS-based genetic testing remains the data analysis, which includes several variants of unknown significance. Functional tests are therefore crucial to assess the pathogenicity of new variants that are detected by NGS.

One of the major advantages of the NGS approach is the identification of both polygenic conditions and modifier variants associated with causative mutations. Indeed, studies of Mendelian conditions have revealed the extent to which many rare diseases can be characterized by complex modes of inheritance, such as digenic inheritance and dual molecular diagnoses, which occur when pathogenic variations at two or more loci lead to expression of two or more Mendelian conditions [10].

In this study, we evaluated a large case series of 155 consecutive patients with different forms of hereditary anemias and erythrocytosis who were referred to the Medical Genetics Unit (‘Federico II’ University Hospital, Naples, Italy) for NGS-based genetic testing, from January 2018 to September 2020. Among the diagnosed patients, 15% showed multi-locus inheritance, which mainly involved *PIEZO1* and *SPTA1* variants.

## 2. Materials and Methods

### 2.1. Patients and Genomic DNA Preparation

In total, 155 patients with clinical suspicion of different types of hereditary anemia were included in this study. Their diagnoses were based on history, clinical findings, and laboratory data. For HS/hereditary stomatocytosis patients, the diagnosis was also based on ektacytometry.

The local University Ethical Committee approved the collection of the patient data (DAIMedLab, ‘Federico II’ University of Naples; N° 252/18). DNA samples were obtained from the patients after they had signed their informed consent, and according to the Declaration of Helsinki. Whenever possible, affected and unaffected relatives were also enrolled to correctly assess the pathogenicity of each variant by analysis of the family segregation.

Genomic DNA preparation was performed as previously described [11]. To evaluate the quality of the extracted genomic DNA before fragmentation, samples were quantified using a UV-Vis spectrophotometer (NanoDrop 2000; Thermo Scientific, Waltham, MA, USA). Then, the genomic DNA was run on 0.8% agarose DNA gel electrophoresis.

### 2.2. Libraries Establishment

Genetic testing was achieved by targeted NGS using a custom 86-gene panel for hereditary RBC defects [12]. This panel is an updated version of a similar previously published panel [13], and it was composed of 86 genes that are causative of congenital dyserythropoietic anemias, Diamond–Blackfan anemia, RBC membrane defects, hemolytic anemias due to RBC enzyme defects, anemias due to iron metabolism defects, hereditary hemochromatosis, and hereditary erythrocytosis.

For the probe design, coding regions, 5ʹUTR, 3ʹUTR, and 50-bp flanking splice junctions were selected as the regions of interest. The probe design was performed using the web-based tool SureDesign (https://earray.chem.agilent.com/suredesign.htm, accessed on 2 August 2019; Agilent Technologies, Santa Clara, CA, USA). The sequence length was set at 150 × 2 nucleotides. The total probe size was 298,393 kbp. Sample preparation was performed using target enrichment (SureSelectQXT) for the Illumina platform (SureSelect Custom Tier1 1–499 kb; Agilent Technologies, Santa Clara, CA, USA), according to the manufacturer’s instructions.

### 2.3. Sequencing and Data Analysis

High-throughput sequencing was performed using a benchtop sequencer (MiSeq; Illumina). The alignment of sequencing reads to the genomic locations, quality control metrics, and identification of variants were achieved using the Alissa Align and Call software (v1.1.2-2; Agilent Technologies, Santa Clara, CA, USA). Variant annotation and analysis were performed using the Alissa Interpret software (v5.2.6; Agilent Technologies, Santa Clara, CA, USA). As previously described and according to the guidelines of the American College of Medical Genetics and Genomics (ACMG), the pathogenicity of each variant was evaluated by gathering evidence from various sources: population data, computational and predictive data, functional data, and segregation data [13].

Due to the large range of prevalence in the population of these heterogeneous disorders, we selected both rare and low-frequency variants (minor allele frequency: <0.01, 0.05, respectively), as reported by the gnomAD browser (https://gnomad.broadinstitute.org/ accessed on 5 April 2021). The InterVar (http://wintervar.wglab.org/ accessed on 5 April 2021) and Varsome (https://varsome.com/ accessed on 5 April 2021) web tools were used for clinical interpretation of the new variants, following the ACMG and the Association for Molecular Pathology guidelines [14]. Automated output was adjusted using the available evidence for each patient (Table 1 and Appendix A). For the functional data criteria for the new variants, we did not perform in vitro functional studies that are supportive of a damaging effect on the gene or gene product, as needed to obtain strong evidence of pathogenicity. However, we selected and report those variants with moderate pathogenic evidence, i.e., variants located in a mutational hot spot and/or critical and well-established functional domain. Moreover, the validation of the variants was assessed by analysis of clinical data and family history, ektacytometry analysis, and peripheral blood evaluation, whenever possible.

All of the prioritized variants were confirmed by Sanger sequencing and by analysis of inheritance patterns, whenever possible. The validations were performed using 50 ng genomic DNA. Custom primers were designed using the Primer3 software (v. 0.4.0; freeware online). The primer sequences are available on request (roberta.russo@unina.it). Nucleotide numbering reflects cDNA numbering with +1 corresponding to the ‘A’ of the ATG translation initiation codon in the reference sequence, according to the nomenclature for the description of sequence variants of the Human Genome Variation Society (www.hgvs.org/mutnomen, accessed on 5 April 2021). The initiation codon is codon 1.

### 2.4. Gene Ranking

To rank the mutated genes based on their genic intolerance, we used the Residual Variation Intolerance Score (RVIS) percentile, as retrieved from the Genic Intolerance database (http://genic-intolerance.org/, accessed on 5 April 2021). The RVIS is a gene-based score that is designed to rank genes in terms of whether they have more or less common functional genetic variation relative to the genome-wide expectation, given the amount of apparently neutral variation the gene has. A gene with a positive score has more common functional variation, and a gene with a negative score has less, and is referred to as ‘intolerant’ [15].

### 2.5. Statistical Analysis

Quantitative data were compared using Mann–Whitney tests. Multiple comparisons were performed using Kruskal–Wallis tests, with post-hoc correction using Dunn’s multiple comparison tests. Qualitative data were compared using chi-squared tests. A two-sided *p* < 0.05 was considered as statistically significant.

## 3. Results

### 3.1. NGS-Based Genetic Testing for Identification of Multiple Disease-Causing Genotypes

Among the 155 patients originally suspected of red blood cell defects, final diagnoses were reached for 130/155 (84%). Overall, 69% (107/155) showed monogenic inheritance, and 15% (23/155) showed multi-locus inheritance (Figure 1A). The genetic features of the patients with multi-locus inheritance patterns are summarized in Table 2. The complete clinical features were available for only 20/23 patients within this subset (Appendix A).

Among the patients with multi-locus molecular diagnosis, 18/23 (78%) showed variants of the *PIEZO1* gene, and 7/23 (30%) showed SPTA1 variants. In agreement with this, *PIEZO1* and *SPTA1* were the most mutated loci among the other causative genes identified in this case series (Figure 1B). Of note, the high frequencies of mutations in both of these genes were mainly related to their low genic intolerance, as suggested by the high values of the RVIS percentiles for both genes (Figure 1B). Accordingly, most of the variants in *PIEZO1* and *SPTA1* genes were originally predicted as variants of uncertain significance (VUS) or likely benign (Table 1). Interestingly, the reevaluation of *PIEZO1* and *SPTA1* pathogenic variants by ACMG rules demonstrated that 26/35 (74%) and 17/35 (48.6%) *PIEZO1* variants were predicted as VUS by InterVar and Varsome tools, respectively. Similarly, *SPTA1* pathogenic variants were predicted as VUS in a range from 43.8% (Varsome) to 93.8% (InterVar) (Appendix A). Accordingly, whenever possible, automated outputs were adjusted and explained using the available evidence for each patient (Table 1 and Appendix A).

### 3.2. Blood Count, Hemolytic Markers, and Iron Balance of Dual Inheritance Patients

In total, 16 of the 23 patients with multi-locus diagnosis (70%) showed dual molecular diagnosis of hereditary stomatocytosis, most of which were affected by dehydrated hereditary stomatocytosis type 1 (DHS1), and hereditary spherocytosis mainly due to biallelic SPTA1 variants (Table 2). Here, we named this combination as “dual inheritance”. The family segregation of the variants identified in some of the patients with multi-locus inheritance is summarized in Appendix A.

To investigate the multi-locus contributions to the hematological phenotype in these dual inheritance patients (only those patients with dual inheritance of DHS and HS (*n* = 16) with code P1, P3-P9, P11-P15, P18_P20, indicated in bold in Table 2), some RBC indices were compared with those with DHS1 (*n* = 37, patients with clinical and molecular diagnosis described in [16]) and HS (*n* = 21, patients with clinical and molecular diagnosis included in the 155 ones here described): hemoglobin (Hb), mean corpuscular volume [MCV], mean corpuscular hemoglobin [MCH], and ferritin:age ratio (Figure 2A and Table 3). Overall, there were no differences in these RBC indices between dual inheritance and HS, while significant differences were seen for dual inheritance versus DHS1 (Figure 2A and Table 3). Indeed, dual inheritance patients showed lower Hb, MCV, and MCH compared to DHS1 patients. As expected, MCV and MCH were higher in DHS1 patients than HS patients (Figure 2A and Table 3). Interestingly, DHS1 patients showed significantly higher ferritin:age ratio compared to both dual inheritance and HS patients (Figure 2A and Table 3).

### 3.3. Hydration and Deformability Status of Dual Inheritance Patients

To investigate the multi-locus contributions to the hydration and deformability of RBCs of some representative dual inheritance patients (*n* = 11), the ektacytometry curves were analyzed in comparison with DHS1 (*n* = 18, patients with clinical and molecular diagnosis described in [16]) and HS (*n* = 116, on the basis of the recently published study [17]). Some of these Osmoscan profiles for the dual inheritance patients are reported in Appendix A. Three Osmoscan parameters were evaluated across the three subgroups of patients: (i) the minimum osmolality of ‘O min’, as the osmolality at which the deformability reaches a minimum, which represents 50% of RBC hemolysis in conventional osmotic fragility assays, and reflects the mean cell surface:volume ratio; (ii) the maximum elongation index as ‘EI max’, which corresponds to the maximal deformability or elongation obtained near the isotonic osmolality, and is an expression of the membrane surface; and (iii) the hypertonic osmolality as ‘O hyper’, which represents the osmolality in the hypertonic region that corresponds to 50% of the EI max, and which reflects the mean cell hydration status. Overall, there were no significant differences in ‘Omin’ and ‘EI max’ between dual inheritance and DHS1 (Figure 2B). Indeed, these analyses demonstrated lower O min and O hyper for dual inheritance compared to HS, while EI max was significantly higher in dual inheritance compared to HS (Figure 2B). The parameter ‘O hyper’ was decreased in dual inheritance compared to both DHS1 and HS demonstrating a substantial dehydration status of RBCs (Figure 2B).

## 4. Discussion

Multi-locus inheritance defines a genetic disease that arises from mutations to more than one gene. Such multi-locus cases can be caused by (i) biallelic or triallelic mutations in two distinct genes, *in cis* or *trans*; (ii) co-inheritance of pathogenic variants responsible for two or more distinct disease entities, which can lead to a mixed phenotype; or (iii) pseudo-multi-locus inheritance due to monogenic Mendelian conditions, with a broad spectrum of phenotypes due to co-inheritance of genetic modifiers [18]. It is now clear that many Mendelian conditions can be characterized by complex modes of inheritance. In this context, NGS-based genetic testing has revolutionized the diagnosis of genetic diseases by identification of multi-locus inheritance for several diseases, such as Charcot–Marie–Tooth disease [19], polycystic kidney disease [20], and congenital hypogonadotropic hypogonadism [21].

In the present study, we examined multi-locus inheritance in patients with RBC defects. A case series of 155 consecutive patients were examined who were referred in the Medical Genetics Unit (‘Federico II’ University Hospital, Naples, Italy) from January 2018 to September 2020. The genetic analysis was conducted using a third version of a custom targeted NGS panel that included 86 causative genes of hereditary RBC defects. We obtained an overall diagnostic yield of 84% of the patients, thus improving the diagnostic yield previously reached by the first (34 genes) and second (71 genes) versions of these NGS panels targeted for RBC defects (65%) [13,22]. This is in agreement with diagnostic rates reported in the literature. Indeed, the custom panels for hereditary anemias that are available include variable numbers of genes (e.g., 50–200) with diagnostic rates reported from 38% to 87%, which depend on how many and which genes are included, and on the depth of the phenotypic assessment required [9]. Our diagnostic workflow provides the use of whole exome sequencing in the negative cases to find new causative genes of RBC defects, or of CGH-array to find possible deletion/duplication.

In the present cohort of patients, 69% showed monogenic inheritance and 15% multi-locus inheritance, as mainly a dual molecular diagnosis. These data are in agreement with the literature. Indeed, dual molecular diagnoses have been reported to account for at least 4% of diagnosed cases [23], with higher rates seen for case series of selected phenotypes (12%) [24] and for cases with apparent phenotypic expansion (32%) [25].

Here, most of these dual inheritance cases were due to co-inheritance of pathogenic variants of the *PIEZO1* and *SPTA1* genes, which are responsible for two distinct diseases: DHS1 and HS. Accordingly, these genes were the most frequently mutated loci among the causative genes identified in the present case series. *PIEZO1* and *SPTA1* are two large and highly polymorphic genes that show reduced genetic constraints. Of note, our data suggested that ACMG classification fails to assess the pathogenicity of genetic variants in both genes. Indeed, although the ACMG guidelines were intended to be used universally for all Mendelian disorders, certain criteria require gene- or disease-specific knowledge for an accurate variant interpretation [26]. The mutated genes identified here were classified based on their genic intolerance, using RVIS percentile scores. According to their low intolerance to variation, the *PIEZO1* and *SPTA1* genes showed the highest RVIS values. Indeed, the more intolerant to variation a gene is, the less likely it is to be mutated. Interestingly, among the causative genes described here, *PKLR* and *SEC23B* also showed high frequencies of mutations, and accordingly, they showed intermediate RVIS percentiles. This might explain the high prevalence of patients with congenital dyserythropoietic anemia II among those with congenital dyserythropoietic anemias, as well as the increased number of patients diagnosed with pyruvate kinase deficiency in recent years [3].

The multi-locus inheritance led to a mixed phenotype that was more similar to HS than DHS1, in terms of the RBC indices of Hb, MCV, and MCH. Interestingly, as assessed by the ferritin:age ratio, the iron balance of these dual inheritance cases was also more similar to HS than DHS1. Therefore, in terms of the Hb levels, the clinical phenotype was more severe for these dual inheritance patients compared to DHS1. In contrast, for MCV, MCH, and the ferritin:age ratio, the phenotype of the multi-locus cases was milder compared to DHS1.

Within the present case series, there were seven severe cases who were transfusion dependent (patients P2, P3, P4, P6, P7, P10, P20). Of note, four of these seven transfusion-dependent patients showed dual inheritance of DHS1/HS. The hemolytic indices for bilirubin and LDH did not differ in the multi-locus patients compared to those who were monogenic. However, most of the multi-locus patients had splenomegaly, and four of them had undergone splenectomy.

Here, we also identified a rare case of a syndrome characterized by hydrops, lactic acidosis, and sideroblastic anemia, which was due to a homozygous variant in the *LARS2* gene. This patient (P8) was a 3-year-old Swedish female who showed multi-locus inheritance of variants in the *ABCB6*, *KCNN4*, and *LARS2* genes. Indeed, she had microcytic anemia with fetal ascites that required two intrauterine transfusions. She also showed impairment of psychomotor development and bilateral deafness.

We further characterized the deformability and hydration status of RBCs from the patients with dual inheritance by ektacytometry analysis. Here, the co-inheritance of DHS1 and HS resulted in peculiar bell-shaped ektacytometry curves that were left shifted, as for the patients with DHS1. Indeed, comparisons of the ‘O hyper’ parameter (which reflects the hydration status) among these dual inheritances, DHS1, and HS patients highlighted lower values for dual inheritance compared to both DHS1 and HS. Overall, comparisons of these ektacytometry curves for dual inheritance with those of DHS1 and HS highlighted the dehydrated conditions of the RBCs, as seen for DHS1. The ‘O min’ value, which reflects the mean cell surface:volume ratio, was more similar to DHS1. Finally, the ‘EI max’ (as an expression of the cell membrane surface) was higher for the multi-locus cases compared to both DHS1 and HS.

## 5. Conclusions

In this study, we evaluated a large case series of 155 consecutive patients with different forms of RBC defects who were referred to the Medical Genetics Unit (‘Federico II’ University Hospital, Naples, Italy) for NGS-based genetic testing, from January 2018 to September 2020. Among the diagnosed patients, 15% showed multi-locus inheritance, which mainly involved *PIEZO1* and *SPTA1* variants. The data from the present study demonstrate that the first and second line of investigations included in the conventional workflow for diagnosis of hereditary anemias can fail to provide differential diagnoses for patients with multi-locus inheritance. Indeed, we have shown that the clinical parameters, such as the RBC indices and the iron status, are not informative for any differential diagnosis of dual inherited conditions. Moreover, the Osmoscan profile did not provide easy discrimination between these multi-locus and monogenic RBC defects.

Of note, our study further highlighted the importance to reevaluate the pathogenicity of the identified genetic variants in light of the new data presented in the literature and of the follow-up of the clinical case. For *PIEZO1* and *SPTA1* genes, we demonstrated that the ACMG rules often failed to assess the pathogenicity of the identified variants. For this reason, the introduction of functional tests is useful to define the pathogenicity of VUS and to establish a correct genotype–phenotype relationship.

Genetic testing is already a routine part of the diagnostic workflow for patients with RBC defects, and is indeed widespread in clinical practice. Additionally, genetic testing is used more early in the diagnostic workflow of hereditary anemias [8,9]. These data further demonstrate the crucial role for NGS-based genetic testing for diagnosis of such RBC defects, and also for the identification of multi-locus inheritance. Correct genetic diagnosis has become important also to guide treatment and personalized clinical management of these patients. For example, for patients with multi-locus inheritance caused by DHS1 and HS, it is crucial to avoid splenectomy, which is beneficial for patients with HS but contraindicated for DHS1.

## Figures and Tables

**Figure 1 genes-12-00958-f001:**
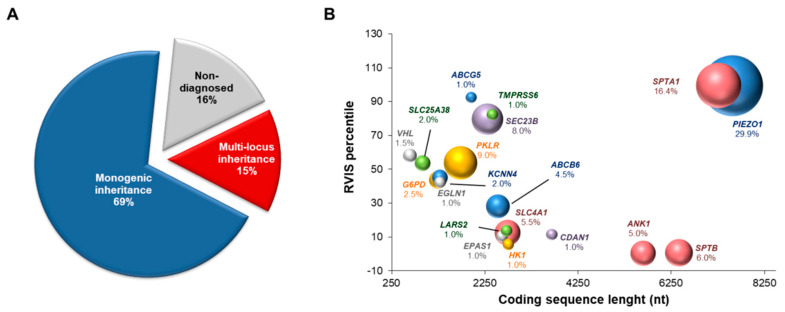
Molecular features of patients with hereditary red blood cell disorders. (**A**) Pie chart showing the proportions of patients diagnosed as monogenic (single gene condition) and multi-locus diseases. The undiagnosed cases evaluated by extended targeted next-generation sequencing for hereditary anemias are also shown. (**B**) Bubble chart defining the lengths of the coding sequences of each hereditary anemia causative gene and their relative Residual Variation Intolerance Score (RVIS) percentiles. Low RVIS percentiles identify increased constraints (intolerance to variation). The size of each bubble represents the frequency of the mutations in each gene, as calculated by the ratio of the number of mutated alleles for each gene and the overall count of disease alleles (*n* = 207).

**Figure 2 genes-12-00958-f002:**
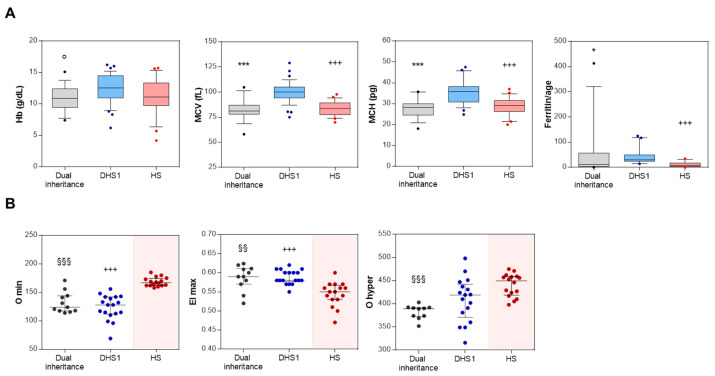
Hematological features and Osmoscan indices of digenic/oligogenic patients. (**A**) Hemoglobin (Hb) (*n* = 16), mean corpuscular volume (MCV) (*n* = 16), mean corpuscular hemoglobin (MCH) (*n* = 16), and ferritin/age levels in patients with dual inheritance (*n* = 16, patients indicated in bold in Table 2), dehydrated hereditary stomatocytosis type 1 (DHS1; *n* = 37), and hereditary spherocytosis (HS; *n* = 21). Data are medians and whiskers for 10-90 percentiles. (**B**). O min, EI max, and O hyper values from Osmoscan profiles of patients with dual inheritance (*n* = 11), DHS1 (*n* = 18), and HS (*n* = 16). Data are medians and interquartile range. °, *p* <0.05 (dual inheritance vs. DHS1, Student’s *t* test). *, *p* < 0.05; ***, *p* < 0.0001 (dual inheritance vs. DHS1). +++, *p* < 0.0001 (HS vs. DHS1). §§, *p* < 0.001 (dual inheritance vs. HS). §§§, *p* < 0.0001 (dual inheritance vs. HS) (Kruskal–Wallis tests, with post-hoc correction for internal comparisons by Dunn’s multiple comparison tests).

**Table 1 genes-12-00958-t001:** Variant classification and reassessment.

Gene	HGVS Nomenclature		ACMG Rules ^†^																																	Method	Class
	cDNA-level	Protein-level	PVS1	PS1	PS2	PS3	PS4	**PS5**	PM1	PM2	PM3	PM4	PM5	PM6	**PM7**	PP1	PP2	PP3	PP4	PP5	**PP6**	BA1	BS1	BS2	BS3	BS4	**BS5**	BP1	BP2	BP3	BP4	BP5	BP6	BP7	**BP8**		
*ABCB6*	c.1361T>C	p.Val454Ala																																		Automated	B
																																				Adjusted	LP
*ABCB6*	c.1402G>T	p.Ala468Ser																																		Automated	V
																																				Adjusted	LP
*ABCB6*	c.1474G>A	p.Ala492Thr																																		Automated	B
																																				Adjusted	LP
*ABCB6*	c.1691T>C	p.Met564Thr																																		Automated	V
																																				Adjusted	V
*ABCB6*	c.1762G>A	p.Gly588Ser																																		Automated	B
																																				Adjusted	V
*ABCB6*	c.2215C>T	p.Arg739Cys																																		Automated	V
																																				Adjusted	LP
*ANK1*	c.613-1G>C	-																																		Automated	P
																																				-	-
*ANK1*	c.1540G>T	p.Gly514Cys																																		Automated	V
																																				Adjusted	LP
*G6PD*	c.1360C>T	p.Arg454Cys																																		Automated	P
																																				-	-
*KCNN4*	c.983A>G	p.His328Arg																																		Automated	V
																																				Adjusted	LP
*KCNN4*	c.1018C>A	p.His340Asn																																		Automated	V
																																				Adjusted	LP
*LARS2*	c.457A>C	p.Asn153His																																		Automated	V
																																				Adjusted	LP
*PIEZO1*	c.608T>C	p.Leu203Pro																																		Automated	LB
																																				Adjusted	V
*PIEZO1*	c.1001C>T	p.Ala334Val																																		Automated	V
																																				Adjusted	LP
*PIEZO1*	c.1447G>A	p.Val483Met																																		Automated	V
																																				Adjusted	V
*PIEZO1*	c.1813A>G	p.Met605Val																																		Automated	V
																																				Adjusted	LP
*PIEZO1*	c.3935C>T	p.Ala1312Val																																		Automated	B
																																				Adjusted	LP
*PIEZO1*	c.4481A>C	p.Glu1494Ala																																		Automated	V
																																				Adjusted	LP
*PIEZO1*	c.5195C>T	p.Thr1732Met																																		Automated	B
																																				Adjusted	V
*PIEZO1*	c.5835C>G	p.Phe1945Leu																																		Automated	V
																																				Adjusted	LP
*PIEZO1*	c.5981C>G	p.Ser1994Cys																																		Automated	V
																																				Adjusted	LP
*PIEZO1*	c.6205G>A	p.Val2069Met																																		Automated	V
																																				Adjusted	LP
*PIEZO1*	c.6796G>A	p.Val2266Ile																																		Automated	V
																																				Adjusted	P
*PIEZO1*	c.7180G>A	p.Gly2394Ser																																		Automated	LB
																																				Adjusted	P
*PIEZO1*	c.7219G>C	p.Glu2407Gln																																		Automated	V
																																				Adjusted	P
*PIEZO1*	c.7367G>A	p.Arg2456His																																		Automated	LP
																																				Adjusted	P
*PIEZO1*	c.7529C>T	p.Pro2510Leu																																		Automated	B
																																				Adjusted	LP
*PIEZO1*	c.7558A>G	p.Lys2520Glu																																		Automated	B
																																				Adjusted	LP
*PKLR*	c.1675C>T	p.Arg559*																																		Automated	P
																																				-	-
*SEC23B*	c.1233+4C>T	-																																		Automated	V
																																				Adjusted	V
*SLC4A1*	c.1462G>A	p.Val488Met																																		Automated	P
																																				-	-
*SLC4A1*	c.2608C>T	p.Arg870Trp																																		Automated	V
																																				Adjusted	LP
*SLC4A1*	c.2621T>C	p.Leu874Pro																																		Automated	V
																																				Adjusted	V
*SPTA1*	c.460_462dupTTG	p.Leu155dup																																		Automated	V
																																				Adjusted	P
*SPTA1*	c.1958A>G	p.Tyr653Cys																																		Automated	B
																																				Adjusted	LP
*SPTA1*	c.2173C>T	p.Arg725*																																		Automated	P
																																				-	-
*SPTA1*	c.2464+1G>A	-																																		Automated	P
																																				-	-
*SPTA1*	c.4708G>A	p.Ala1570Thr																																		Automated	V
																																				Adjusted	V
*SPTA1*	c.5029G>A	p.Gly1677Arg																																		Automated	V
																																				Adjusted	LP
*SPTA1*	c.5183G>A	p.Trp1728*																																		Automated	P
																																				-	-
*SPTB*	c.40C>T	p.Pro14Ser																																		Automated	V
																																				Adjusted	V
*SPTB*	c.871G>A	p.Gly291Ser																																		Automated	V
																																				Adjusted	LP
*SPTB*	c.1606G>A	p.Asp536Asn																																		Automated	B
																																				Adjusted	V

^†^ Additional evidence is highlighted in bold (http://wintervar.wglab.org/, accessed on 5 April 2021). NCBI RefSeq transcript for each gene: *ABCB6*, NM_005689; *ANK1*, NM_000037; *G6PD*; NM_001042351; *KCNN4*, NM_002250; *LARS2*, NM_015340; *PIEZO1*, NM_001142864; *PKLR*, NM_000298; *SEC23B*, NM_006363; *SLC4A1*, NM_000342; *SPTA1*, NM_003126; *SPTB*, NM_001355437. The relative values of each cell are represented by a color scale: red cells indicate evidence of pathogenicity while green cells indicate evidence of non-pathogenicity. The color scale identifies the variant classification: dark red, pathogenic (P); red, likely pathogenic (LP); yellow, variants of unknown significance (VUS); light green, likely benign (LB); green, benign (B).

**Table 2 genes-12-00958-t002:** Genetic features of the patients with multi-locus inheritance.

Patient ID	Disease	Gene	HGVS Nomenclature		Zygosity	RefSeq ID	AF	HGMD ID
			cDNA-Level	Protein-Level			gnomAD ^§^	
**P1**	**HS/FP**	*ABCB6*	NM_005689:c.C2215T	p.Arg739Cys	Het	rs141840760	0.0004	-
		*SPTA1*	NM_003126:c.5183G>A	p.Trp1728*	Comp het	-	-	-
		*SPTA1*	NM_003126:c.6531-12C>T	-	Comp het	rs28525570	0.23	CS995155
P2	HS/CDAII	*SEC23B*	NM_006363:c.1233+4C>T	-	Hom	rs201883785	-	-
		*SPTA1*	NM_003126:c.4708G>A	p.Ala1570Thr	Hom	rs778626016	-	-
		*SPTA1*	NM_003126:c.6531-12C>T	-	Hom	rs28525570	0.23	CS995155
**P3**	**DHS1/HS**	*SPTA1*	NM_003126:c.6531-12C>T	-	Comp het	rs28525570	0.23	CS995155
		*SPTA1*	NM_003126:c.5029G>A	p.Gly1677Arg	Comp het	rs771033064	0	CM187374
		*PIEZO1*	NM_001142864:c.7558A>G	p.Lys2520Glu	Het	rs570744198	0.001	CM187408
**P4**	**DHS1/HE**	*SPTB*	NM_001355437:c.871G>A	p.Gly291Ser	Het	rs143599352	0.0002	-
		*PIEZO1*	NM_001142864:c.7219G>C	p.Glu2407Gln	Het	rs200291894	0.0001	CM1922287
**P5**	**DHS1/HE**	*SPTB*	NM_001355437:c.40C>T	p.Pro14Ser	Het	rs147059670	0.0001	-
		*PIEZO1*	NM_001142864:c.7180G>A	p.Gly2394Ser	Het	rs201950081	0.0001	CM187364
**P6**	**DHS1/HS**	*PIEZO1*	NM_001142864:c.608T>C	p.Leu203Pro	Het	rs977249154	0	-
		*SPTA1*	NM_003126:c.6531-12C>T	-	Het	rs28525570	0.23	CS995155
		*SPTA1*	NM_003126:c.1958A>G	p.Tyr653Cys	Het	rs148912436	0.008	CM187425
**P7**	**DHS1/HS**	*PIEZO1*	NM_001142864:c.4481A>C	p.Glu1494Ala	Het	-	-	-
		*SPTA1*	NM_003126:c.2464+1G>A	-	Hom	rs774632615	0	-
**P8**	**DHS2/FP/HLASA**	*ABCB6*	NM_005689:c.1402G>T	p.Ala468Ser	Het	rs777270402	0	-
		*LARS2*	NM_015340:c.457A>C	p.Asn153His	Hom	rs786205560	-	CM1615275
		*KCNN4*	NM_002250:c.1018C>A	p.His340Asn	Het	rs76935412	0.002	-
**P9**	**HS/FP**	*ABCB6*	NM_005689:c.1762G>A	p.Gly588Ser	Het	rs145526996	0.004	CM128905
		*SLC4A1*	NM_000342:c.2621T>C	p.Leu874Pro	Het	-	-	-
P10	PKD/DHS1	*PKLR*	NM_000298:c.1675C>T	p.Arg559*	Hom	rs532230312	0	CM981585
		*PIEZO1*	NM_001142864:c.6796G>A	p.Val2266Ile	Het	rs546338962	0	CM187363
**P11**	**DHS1/HE**	*PIEZO1*	NM_001142864:c.5195C>T	p.Thr1732Met	Het	rs139051768	0.011	-
		*SPTB*	NM_001024858:c.1606G>A	p.Asp536Asn	Het	rs145675502	0.001	CM187385
**P12**	**DHS1/FP**	*PIEZO1*	NM_001142864:c.7367G>A	p.Arg2456His	Het	rs587776988	-	CM127746
		*ABCB6*	NM_005689:c.1474G>A	p.Ala492Thr	Het	rs147445258	0.007	CM169662
**P13**	**DHS1/FP**	*ABCB6*	NM_005689:c.1361T>C	p.Val454Ala	Het	rs61733629	0.006	CM169864
		*PIEZO1*	NM_001142864:c.5981C>G	p.Ser1994Cys	Het	-	-	-
**P14**	**DHS1/HS**	*SLC4A1*	NM_000342:c.1462G>A	p.Val488Met	Het	rs28931584	0	CM971385
		*PIEZO1*	NM_001142864:c.5195C>T	p.Thr1732Met	Het	rs139051768	0.011	CM200163
**P15**	**DHS1/HS**	*SLC4A1*	NM_000342:c.2608C>T	p.Arg870Trp	Het	rs28931585	-	CM951173
		*PIEZO1*	NM_001142864:c.1001C>T	p.Ala334Val	Het	rs574402639	0.0003	-
P16	DHS2/HS	*KCNN4*	NM_002250: c.1018C>A	p.His340Asn	Het	rs76935412	0.002	-
		*SPTA1*	NM_003126:c.460_462dupTTG	p.Leu155dup	Het	rs757679761	0	CI890173
		*SPTA1*	NM_003126:c.6531-12C>T	-	Het	rs28525570	0.23	CS995155
P17	DHS1/DHS2	*PIEZO1*	NM_001142864:c.1813A>G	p.Met605Val	Het	rs1490094586	-	CM1911810
		*KCNN4*	NM_002250:c.983A>G	p.His328Arg	Het	rs780167756	-	-
**P18**	**DHS1/HS**	*ANK1*	NM_020476:c.613-1G>C	-	Het	-	-	-
		*PIEZO1*	NM_001142864:c.3935C>T	p.Ala1312Val	Het	rs34246477	0.0014	-
**P19**	**DHS1/HS**	*PIEZO1*	NM_001142864:c.3935C>T	p.Ala1312Val	Het	rs34246477	0.001	-
		*ANK1*	NM_020476:c.613-1G>C	-	Het	-	-	-
**P20**	**DHS1/HS**	*PIEZO1*	NM_001355436:c.7529C>T	p.Pro2510Leu	Het	rs61745086	0.007	CM1812923
		*ANK1*	NM_000037:c.1540G>T	p.Gly514Cys	Het	rs199975878	0	-
P21	DHS1/HS	*PIEZO1*	NM_001142864:c.1447G>A	p.Val483Met	Het	rs747301309	0	-
		*SPTA1*	NM_003126:c.2173C>T	p.Arg725*	Het	-	-	-
		*SPTA1*	NM_003126:c.2909C>A	p.Ala970Asp	Het	rs35948326	0.03	CM930690
P22	DHS1/FP	*PIEZO1*	NM_001142864:c.5835C>G	p.Phe1945Leu	Het	rs776602133	0	-
		*ABCB6*	NM_005689:c.1691T>C	p.Met564Thr	Het	rs1233572695	-	-
P23	DHS1/G6PD	*PIEZO1*	NM_001142864:c.6205G>A	p.Val2069Met	Het	rs199752762	0.001	-
		*G6PD*	NM_001042351:c.1360C>T	p.Arg454Cys	Hem	rs398123546	0.0001	CM920290

^§^ AF, alternative allele frequency, as reported in the gnomAD browser (https://gnomad.broadinstitute.org/, accessed on 5 April 2021); CDAII, congenital dyserythropoietic anemia type II; DHS1, dehydrated hereditary stomatocytosis type 1; DHS2, dehydrated hereditary stomatocytosis type 2; HE, hereditary elliptocytosis; HS, hereditary spherocytosis; HLSA, hydrops, lactic acidosis, and sideroblastic anemia/Perrault syndrome; FP, familial pseudohyperkalemia; G6PD, hemolytic anemia, G6PD deficient (favism); PKD, pyruvate kinase deficiency. Het, heterozygous; Hom, homozygous; Comp het, compound heterozygous; HGVS, Human Genome Variation Society database; HGMD, Human Gene Mutation database (HGMD Professional 2020.3). In bold are indicated the patients analyzed in Figure 2.

**Table 3 genes-12-00958-t003:** Clinical features of the case series enrolled in the study.

Analysis	Unit	DHS1 Cases	HS Cases	Dual Inheritance	Reference Range	P ^§^	P^1^	P^2^	P^3^
		(*n* = 37)	(*n* = 21)	(*n* = 16)					
Age	years	20.5 ± 3.2 (20.0; 28)	27.8 ± 4.0 (25.0; 20)	23.9 ± 5.1 (18.5; 16)	-	0.32	-	-	-
Gender	female/male	17 (45.9)/20 (54.1)	16 (76.2)/5 (23.8)	5 (29.4)/11 (70.6)	-	0.02	0.32	<0.01	0.03
**Hematological data**									
RBCs	× 10^6^/µL	3.1 ± 0.1 (3.0; 36)	3.9 ± 0.2 (4.3; 19)	4.0 ± 0.2 (3.9; 15)	4.0–5.2	<0.001	0.004	1.00	0.004
Hb	g/dL	12.5 ± 0.4 (12.7; 36)	11.2 ± 0.6 (11.1; 21)	10.9 ± 0.5 (10.8; 16)	11.5–15.5	0.06	-	-	-
Ht	%	36.3 ± 1.7 (37.3; 36)	33.1 ± 1.6 (33.1; 19)	31.8 ± 1.6 (30.5; 14)	35–45	0.04	0.11	1.00	0.14
MCV	fL	100.5 ± 1.7 (100.0; 36)	83.2 ± 1.6 (83.8; 21)	82.5 ± 2. 5 (80.9; 16)	77–95	<0.001	<0.001	1.00	<0.001
MCH	pg	35.9 ± 1.1 (36.0;27)	28.8 ± 0.9 (29.0; 21)	27.6 ± 1.1 (27.4; 16)	25–33	<0.001	<0.001	1.00	<0.001
MCHC	g/dL	35.5 ± 0.7 (34.3; 36)	34.2 ± 0.9 (34.5; 19)	34.1 ± 0.5 (34.1; 16)	32–36	0.81	-	-	-
PLTs	× 10^3^/µL	414.2 ± 55.3 (375.0; 15)	276.3 ± 34.1 (270.0; 20)	288.7 ± 41.8 (232.0; 16)	150–450	0.05	0.08	1.00	0.09
ARC	× 10^3^/µL	164.5 ± 17.9 (145.9; 33)	238.3 ± 45.9 (234.6; 16)	217.5 ± 50.5 (144.5; 13)	20–90	0.49	-	-	-
**Biochemical data and iron balance**									
Total bilirubin	mg/dL	3.7 ± 0.6 (3.9; 21)	2.9 ± 0.6 (2.4; 16)	3.0 ± 0.5 (3.5; 12)	0.3–1.0	0.59	-	-	-
Ferritin	ng/mL	491.3 ± 82.1 (363.0; 19)	319.1 ± 117.8 (150.2; 14)	409.8 ± 154.4 (128.5; 13)	22–275	0.18	-	-	-
Ferritin level/dosage age ^†^	-	38.9 ± 6.5 (30.3; 18)	10.9 ± 2.9 (7.9; 14)	36.2 ± 16.1 (11.0; 13)	-	<0.001	0.004	1.00	<0.001

DHS1, dehydrated hereditary stomatocytosis type I (*PIEZO1* mutated); HS, hereditary spherocytosis; RBCs, red blood cells; Hb, hemoglobin; Ht, hematocrit; MCV, mean corpuscular volume; MCH, mean corpuscular hemoglobin; MCHC, mean corpuscular hemoglobin concentration; PLTs, platelets; ARC, absolute reticulocyte count; Quantitative variables data are presented as mean ± SEM (median; number of cases [n]); Qualitative variables data are presented as n (%)/n (%); ^§^ Kruskal–Wallis tests for quantitative unpaired data; chi-squared tests for categorical data; P^1^ DHS1 vs. dual inheritance cases; P^2^ HS vs. dual inheritance cases; P^3^ DHS1 vs. HS cases (Bonferroni post-hoc tests for multiple comparisons); ^†^ Normalization of ferritin using “Ferritin level/dosage age ratio”, as described by [16].

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
