# Peer review of "Complex Modes of Inheritance in Hereditary Red Blood Cell Disorders: A Case Series Study of 155 Patients"

_genes, 2021, doi:10.3390/genes12070958_

Round 1

Reviewer 1 Report

This is a clearly written report of an important clinical problem by authors who are noted experts in this area. Occasionally, when NGS panels fail to reveal the molecular bases of a condition with a definite hemolytic phenotype, researchers extent the work-up to exome or whole genome sequencing. I wonder, in the 16% of patients tested where no diagnosis was identified with the 96 gene panel, was such diagnostic extension performed in any? At least a brief comment on this issue would give readers the advice of these experts on what to do when NGS panels are unrevealing but a significant hematological problem persists. 

Author Response

Reply: We thank the reviewer for his/her positive comments. We added a paragraph regarding the management of negative cases to the first genetic analysis (t-NGS panel) in the discussion. Usually, we proceed with a whole-exome analysis to find new causative genes of HA or with CGH-array to find possible deletion/duplication.

Reviewer 2 Report

The article deals with a very important subject, of high clinical relevance. The authors managed to provide evidence for the substantial number of dual inheritance cases in patients with DHS and HS diagnosis. The strong, but not fully used side of this manuscript is the introduction of functional tests that may provide information to establish phenotype-genotype link for these cases that would add to the clinical importance of the genetic findings. 

Author Response

Reply: We thank the reviewer for his/her positive comments. The introduction of functional tests to establish phenotype-genotype relationships could be useful for the variants of uncertain significance. For example, for the PIEZO1 gene it could be useful to perform an ion flux assay to assess the pathogenicity of the variants identified. We added a paragraph in the discussion on this topic.